# A Mini-Review of Anti-Listerial Compounds from Marine Actinobacteria (1990–2023)

**DOI:** 10.3390/antibiotics13040362

**Published:** 2024-04-15

**Authors:** Siyanda S. Ngema, Evelyn Madoroba

**Affiliations:** Department of Biochemistry and Microbiology, University of Zululand, KwaDlangezwa 3886, South Africa; ngemasiyanda05@gmail.com

**Keywords:** anti-*Listeria* compounds, marine actinobacteria, natural antimicrobials, antimicrobial resistance

## Abstract

Among the foodborne illnesses, listeriosis has the third highest case mortality rate (20–30% or higher). Emerging drug-resistant strains of *Listeria monocytogenes*, a causative bacterium of listeriosis, exacerbate the seriousness of this public health concern. Novel anti-Listerial compounds are therefore needed to combat this challenge. In recent years, marine actinobacteria have come to be regarded as a promising source of novel antimicrobials. Hence, our aim was to provide a narrative of the available literature and discuss trends regarding bioprospecting marine actinobacteria for new anti-Listerial compounds. Four databases were searched for the review: Academic Search Ultimate, Google Scholar, ScienceDirect, and South African Thesis and Dissertations. The search was restricted to peer-reviewed full-text manuscripts that discussed marine actinobacteria as a source of antimicrobials and were written in English from 1990 to December 2023. In total, for the past three decades (1990–December 2023), only 23 compounds from marine actinobacteria have been tested for their anti-Listerial potential. Out of the 23 reported compounds, only 2-allyoxyphenol, adipostatins E–G, 4-bromophenol, and ansamycins (seco-geldanamycin B, 4.5-dihydro-17-O-demethylgeldanamycin, and seco-geldanamycin) have been found to possess anti-Listerial activity. Thus, our literature survey reveals the scarcity of published assays testing the anti-Listerial capacity of bioactive compounds sourced from marine actinobacteria during this period.

## 1. Introduction

*Listeria monocytogenes* is a major foodborne pathogen that can cause severe listeriosis in humans [1]. Severe listeriosis may be characterized by meningitis, septicaemia, meningoencephalitis in immunocompromised people, invasive infections in newborns and the elderly, fetal malformations, and serious complications in pregnant women (abortion and stillbirth), with a case mortality rate that can range from 20% to 30% [2,3,4,5]. The hospitalization rate of the infection is more than 95% [1]. Although potentially deadly, the disease is relatively rare, with 0.1 to 10 cases per 1 million people annually, depending on the country and area of the world [3]. In contrast to high socio-economic regions of the world, Africa has a comparatively lower prevalence of severe listeriosis, despite the continent generally having a higher burden of foodborne infections [2,5]. Annually, 23,150 illnesses and 5463 deaths result from listeriosis worldwide [2,5]. However, the African continent contributes to 16% of the prevalence [2,5]. Despite this relatively low prevalence, the largest *L. monocytogenes* outbreak in the world happened in South Africa in 2017–2018, with 1060 reported cases [4,6,7]. The occurrence of an outbreak of this magnitude against the backdrop of a historically low prevalence of the disease in Africa is probably due to specific risk factors, and less awareness of *Listeria* transmission and risk factors along the food value chain [5].

Drug-resistant strains of *Listeria* have been documented [8,9], contributing to an escalation in morbidity and mortality rates associated with listeriosis [10]. Consequently, the pursuit of novel anti-Listerial compounds becomes a crucial strategy in addressing this crisis. Nature serves as the primary source of biotechnologically important molecules applicable across diverse fields [11]. Microorganisms, particularly within the *Actinobacteria* phylum, exhibit a notable proficiency in producing a variety of bioactive compounds due to their unique genetic composition [12,13]. While the extensive bioprospecting of terrestrial actinomycetes has diminished the probability of discovering novel bioactive compounds from this source, the largely unexplored marine environment is promising for identifying new actinomycetes with distinctive bioactive secondary metabolites. This potential is attributed to the harsh physicochemical conditions in the sea, such as high salinity, pressure, and low temperature, which create a conducive environment for the microbial synthesis of structurally and functionally unique molecules [14].

This review aimed to discuss the anti-Listerial compounds obtained from marine actinobacteria from the years 1990 to 2023 in the context of the sources of bioactive compounds, their structures, and antimicrobial activities. Furthermore, the review highlights the challenges and opportunities in the search for new anti-Listerial compounds from marine actinobacteria. Some reviews have summarized the sources, chemical structures, and anti-Listerial activities of bioactive compounds derived from marine actinobacteria [15,16]. In contrast, our review is unique as it focusses on consolidating information specifically and exclusively related to anti-Listerial compounds sourced from marine actinobacteria. By concentrating solely on the reporting of these compounds, our review identifies and emphasizes the gaps and trends observed from 1990 to 2023 in anti-Listerial compounds research. This focused analysis serves to guide future research directions, aiding researchers in pinpointing areas that need deeper exploration. 

This literature review was conducted using four databases: Academic Search Ultimate, Google Scholar, ScienceDirect, and South African Thesis and Dissertations. The literature was accessed through the EBSCOhost research platform. The search terms used were “Marine actinomycetes”, “OR”, “actinobacteria”, “AND”, “antimicrobials”, “OR”, “antibiotics”, “OR”, “anti-listerial”, “AND”, “Listeria”.” Individual genera also replaced “actinomycetes” and “actinobacteria” as search terms. Only English-published full-text and peer-reviewed manuscripts were selected for the review. Those manuscripts discussing anti-Listerial compounds obtained from terrestrial actinobacteria or from plant extracts were excluded (Figure 1).

A limitation of this review is that the literature regarding alternative treatments for listeriosis is outside the scope of our discussion.

## 2. Background

### 2.1. Foodborne Listeriosis

Among the foodborne illnesses, listeriosis has the third highest case mortality rate (20–30% or higher) [17]. Thus, listeriosis is a serious public health concern [3,18]. 

#### 2.1.1. Clinical Features

The clinical signs and symptoms of listeriosis can be differentiated into two categories, namely perinatal (i.e., feto-maternal and neonatal) listeriosis and listeriosis in adults. Perinatal listeriosis involves the infection of the fetus via the placenta. This may lead to abortion, stillbirth, or the baby being born with granulomatosis infantiseptica (a generalized infection) presenting as pyogranulomatous micro-abscesses [18,19,20]. For the mother, the infection may be asymptomatic or may present as a flu-like illness characterized by fatigue, headache, chills, and painful joints and muscles around 2 to 14 days prior to the miscarriage [21,22]. For reasons that are not clear, the mother’s central nervous system (CNS) is rarely infected [20,21,22,23]. In 10–15% of perinatal cases, the aspiration of maternal fluids during delivery may cause late-onset neonatal listeriosis [24,25]. The clinical features may include flu-like symptoms accompanied by meningitis, 1–8 weeks postpartum [24,25].

#### 2.1.2. Listeriosis Outbreaks

Listeriosis was first recognized as a foodborne disease in 1981 [26]. Thereafter, deadly outbreaks of the disease have been recorded worldwide [4,6,27,28,29,30,31,32]. The foods commonly implicated in the outbreaks include ready-to-eat (RTE) foods, unpasteurized milk, dairy products (yogurt, cheese, etc.), raw and unprocessed meats, salads, and fresh produce [3,7,33,34]. However, meat and its products have been responsible for all the major foodborne listeriosis outbreaks worldwide [7]. Table 1 summarizes some of the disease outbreaks associated with various meat types. Between 2017 and 2018, South Africa experienced the largest listeriosis outbreak in the world (Table 1) [4,6,7]. Two hundred and sixteen people (out of 1060 patients that were traced) died, resulting in a case mortality rate of 20.4% [4,31]. Whole-genome sequencing (WGS) was employed to track the source of the outbreak. Ready-to-eat-meat products (mainly polony) were linked to the outbreak [4]. The implicated bacterial strain was confirmed to be *L. monocytogenes* sequence type (ST) 6 [4]. Consequently, products were recalled from retailers across the country and from 15 importing African countries [35], and litigation was initiated against the RTE meat manufacturing company, leading to financial losses [31]. Therefore, beyond just the impacted customers, the aftermaths of these incidents have wider socioeconomic effects [31].

#### 2.1.3. Chemotherapeutic Treatment for Human Listeriosis

The feasible solution for the treatment of listeriosis is antimicrobial chemotherapy [48,49]. Usually, *Listeria* species are refractory to the lethal effects of many antibiotics. This is because the species are able to live and multiply within the host cells, where they remain hidden from the antibiotics in the extracellular fluid [50]. There are a limited number of antibiotics able to penetrate the host cells and reach the cytosol where the *Listeria* species normally reside in host cells [19,50]. Therefore, the antibiotic of choice for the treatment of listeriosis must be able to penetrate the host cell in sufficiently high concentrations for it to be efficacious [51,52,53]. Furthermore, the antibiotic must not undergo significant changes in its pH upon penetration into the host cell, as significant changes would hinder its efficacy. In addition, the antibiotic must bind to and block the penicillin-binding protein 3 (PBP 3) of *Listeria*, which result in a bactericidal effect (PBP 3 is an enzyme that catalyzes the last step of peptidoglycan synthesis) [48,52]. In the case of a pregnant woman, adequate concentrations of the antibiotic must be able to cross the placenta for fetal treatment [22,54]. Thus, the first choice of chemotherapy for treating listeriosis mainly includes the following antibiotics: penicillin G, amoxicillin, and ampicillin [54,55]. These antibiotics penetrate the host cells and exert their bactericidal effects by blocking a number of PBPs [50,52,56]. For synergy, the penicillins are commonly combined with an aminoglycoside, traditionally gentamicin [57]. In the second choice of therapy, trimethoprim is combined with a sulfonamide (e.g., sulfamethoxazole) in co-trimoxazole [56,57,58,59]. Other second-tier therapies include fluoroquinolones, vancomycin, and erythromycin (used in pregnancy) [57]. These second-choice therapies are normally reserved for people who are allergic to penicillin [54,57].

Prior to initiating treatment, it is imperative to conduct an in vitro assessment of the antimicrobial drugs used against the clinical *Listeria* isolates from a patient [60]. This is because there has been an increase in reports of antimicrobial-resistant *L. monocytogenes* strains from diverse sources such as meat and meat products [6,7], humans [9,61], animals [9,18,62], and food processing establishments [18,63]. The first strain of *L. monocytogenes* found to possess acquired antimicrobial resistance was a clinical isolate from France in 1988 [9,64,65]. The strain was multidrug resistant [9,64], resistant to aminoglycosides (erythromycin, and streptomycin), chloramphenicol and tetracyclines [64]. The antimicrobial resistance-encoding genes were found to be conferred by pIP811, a self-transferable 37-Kb plasmid [64]. Antimicrobial resistance has since spread, and is a serious public health challenge [8,9].

#### 2.1.4. Resistance to Quinolones and Fluoroquinolones

Quinolones are a class of antimicrobial compounds that possess a 4-quinolone ring. Nearly all quinolones have a fluorine atom in their structure and are therefore called fluoroqinolones. Members of the quinolones and/or fluoroquinoles family include nalidixic acid, nafloxacin and ciprofloxacin. They exert a bactericidal effect by blocking bacterial DNA gyrase and topoisomerase IV, thereby inhibiting DNA and RNA syntheses [48,66,67]. However, *Listeria* spp. that are resistant to this class of antibiotics have been detected in food-producing animals and humans [68,69,70]. The mechanisms by which *Listeria* spp. resist fluoroquinolones include target gene alterations that include, for example, topoisomerase and gyrase gene mutations in some *Listeria* species [7,48,71]. These species also resist antibiotics via the overexpression of efflux pumps, namely Lde, MdrL, and FepA [7,72,73,74,75]. The pumps export an antimicrobial agent out of the cytosol and thus minimize its bactericidal or bacteriostatic effects [76,77]. Some *Listeria* spp. may harbor plasmid-mediated quinolone resistance (PMQR) genes, making them resistant to quinoline [48].

#### 2.1.5. Resistance to Penicillins and Cephalosporins

Members of the penicillin group include penicillin G, penicillin V, ampicillin, methicillin, and oxacillin, and cephalosporins include the following broad-spectrum antibiotics: cefetamet, cefixime, ceftibuten, ceftazidime, cefdinir, cefpodoxime, cefotaxime, ceftriaxone, and cefuroxime [8,66,78,79]. Both penicillins and cephalosporins contain a β-lactam ring that is essential for their antimicrobial activity, which is the inhibition of bacterial cell wall synthesis [8,48,79]. These antimicrobials generally accomplish this inhibition by binding, through their β-lactam rings, to PBP 3, which inactivates the enzyme [80].

*L. monocytogenes* strains are naturally resistant to broad-spectrum cephalosporins due to the antimicrobials’ low affinity for the PBB 3 of the bacteria [48,52,80,81]. In contrast, the bacterial strains are generally susceptible to penicillins, except for oxacillin [48]. However, some of the strains eventually become antimicrobial resistant, mainly through horizontal gene transfer, gene mutation, and biofilm formation [52,82,83,84,85]. The acquired antimicrobial resistance may involve various mechanisms [52]. For example, the efflux pump (AnrAB) in *L. monocytogenes* has been determined to confer resistance to β-lactam antibiotics (oxacillin, ampicillin, cephalosporins, and others) [52]. Another efflux pump (MdrL) makes *L. monocytogenes* resistant to cefotaxime [86]. According to Luque-Sastre et al. [48] and Srinivasan et al. [87], only the penicillin-binding protein gene (*penA*) has been linked with *L. monocytogenes* resistance to penicillin G.

#### 2.1.6. Resistance to Aminoglycosides

Structurally, all aminoglycosides possess a cyclohexane ring and amino sugars [66]. Members of this class of antibiotics include gentamicin, kanamycin, streptomycin, and neomycin [66]. These antibiotics bind to the bacterial 30S ribosomal subunit, thereby blocking protein synthesis [66]. Aminoglycoside-resistant strains have been reported [9,52,53,57,61,87,88]. These strains normally emerge as a result of obtaining plasmid-borne genes and transposons encoding aminoglycoside-modifying enzymes [89]. The enzymes can be classified as follows: acetyltransferase, adenyltransferase, and phosphotransferase [89]. The gene (*aad6*) coding for 6-*N*-streptomycin adenylyltransferase (a streptomycin-modifying class of enzymes) has been detected in *L. monocytogenes* and *L. innocua* isolates [48,69]. Apart from *aad6*, no other aminoglycoside resistance genes have been identified to date in *Listeria* spp. [48,69].

#### 2.1.7. Resistance to Tetracyclines

The tetracycline family includes naturally occurring antibiotics (tetracycline, chlortetracycline, and others) and semi-synthetic antibiotics (minocycline, doxycycline, etc.) [66]. Similar to aminoglycosides, tetracyclines bind to the bacterial 30S ribosomal subunit. This binding prevents the combination of aminoacyl-tRNA molecules with the A site of the ribosome. This family of antibiotics is a broad-spectrum bacteriostatic class [66]. However, the emergence of tetracycline resistance is the most common phenomenon among *Listeria* species [48,90]. The species obtain most of their resistance-encoding genes from *Enterococcus* and *Streptococcus* species through two types of mobile genetic elements: conjugative plasmids and transposons [48]. Usually, the genes [*tet* (S) and *tet* (L)] are transported by plasmids, whereas the *tet* (M) gene is carried by Tn916 (a conjugative transposon) [48,91]. The other tetracycline resistance-encoding genes found in *Listeria* spp. are *tet* (A) and *tet* (K) [48,90]. The genes *tet* (A), *tet* (K) and *tet* (L) code for proton antiporters that facilitate the efflux mechanism, which confers tetracycline resistance only. Meanwhile, the genes [*tet* (M) and *tet* (S)] code for ribosome protection proteins that confer resistance to both tetracycline and minocycline [48,90,91]. 

#### 2.1.8. Resistance to Trimethoprim

Trimethoprim is a broad-spectrum synthetic antibiotic. It inhibits folic acid production by binding to dihydrofolate reductase (DHFR), an enzyme that catalyzes the conversion of dihydrofolic acid (DFA) to tetrahydrofolic acid during folic acid production. Due to its structural similarity to DFA (a DHFR substrate), trimethoprim competes with DFA for binding on the active site of the enzyme; consequently, the folic acid synthesis pathway is stopped or hampered [66].

Even though *L. monocytogenes* is commonly susceptible to trimethoprim [48], cases of resistance against the antibiotic have been published [57,69,92,93]. Usually, the mechanism of resistance against trimethoprim arises through the synthesis of trimethoprim-resistant DHFR in addition to the susceptible enzyme [90,94]. There are two resistance genes in *Listeria* that facilitate this synthesis: *dfr*D and *dfr*G; these are carried by plasmid pIP823 and transposon Tn6198, respectively [48].

#### 2.1.9. Resistance to Chloramphenicol

Initially, *Streptomycess venezuelae* produced chloramphenicol. However, the chloramphenicol antibiotic is now chemically synthesized. Chloramphenical inhibits peptide elongation by binding to 23S rRNA of the 50S ribosomal subunit. However, this broad-spectrum antibiotic tends to have severe side-effects (e.g., lowering of bone marrow function, which may lead to complications such as aplastic anemia and a low white blood cell count) [66]. Therefore, this antibiotic is only employed when all other antibiotics fail in potentially fatal cases. In *Listeria* spp., resistance to chloramphenicol is encoded by *cat* (type A-8) and *floR* genes. The cat genes code for type A chloramphenicol acetyltransferases (Cat), which are responsible for enzymatic inactivation via the acetylation of the antibiotic [95]. In *L. monocytogenes*, the *floR* gene is linked to the export of florfenicol—a fluorinated derivative of chloramphenicol [87].

#### 2.1.10. Resistance to Macrolides

Macrolide antibiotics include members such as erythromycin, clindamycin, and azithromycin. They are broad-spectrum and normally bacteriostatic antibiotics whose mode of action involves inhibiting protein synthesis by binding to the 23S rRNA of the 50S ribosomal subunit [48,66]. Resistance to macrolide is commonly facilitated by rRNA methylases. These enzymes are encoded by *erm* genes, specifically, *erm* (A), *erm* (B), and *erm* (C) in *Listeria* spp. The enzymes modify the target site (23S rRNA) by methylating its adenine base, consequently blocking the antibiotic from binding to the site [48,96,97].

### 2.2. Drivers of Antimicrobial Resistance

Some of the drivers of antimicrobial resistance include the indiscriminate use of antimicrobials in veterinary medicine for prophylaxis, growth promotion, and disease treatment [1,98]. Furthermore, the treatment of listeriosis using antibiotics in humans results in the selection of resistant *L*. *monocytogenes* strains [99]. Therefore, the emergence of multi-drug-resistant strains of *L*. *monocytogenes* poses a challenge due to potential treatment failure, limited treatment options, extended stays in medical care facilities and even fatalities [10,60,100]. For these reasons, novel anti-Listerial compounds are required.

### 2.3. Actinobacteria as Potential Sources of Anti-Listerial Compounds

Actinobacteria are known to be prolific producers of novel secondary metabolites with applications in diverse industries including the pharmaceutical industry [101,102]. The phylum *Actinobacteria* contains gram-positive bacteria with a high G + C content in general (>50 mol%) except *Tropheryma whipplei.* These bacteria can be filamentous or non-filamentous, and are found both in terrestrial and aquatic environments [103]. They grow via a combination of tip extension and hyphae branching. Their optimum growth conditions include a mesophilic temperature (ranging between 25 and 30 °C) and a pH between 6 and 9. However, some thermophilic strains have been isolated in high temperatures ranging from 50 to 60 °C, and others can grow in acidic conditions (pH 3.5) [104]. Many actinobacteria are of the order *Actinomycetales*. Members of this order are called actinomycetes. The suborders of *Actinomycetales* that show significance in bio-prospecting include *Streptomycineae*, *Micromonospineae*, *Corynebacterineae*, and *Streptosporangineae* [105].

#### 2.3.1. *Streptomycineae*

The suborder *Streptomycineae* has a single family, *Streptomycetaceae*. Family members are often called streptomycetes and play a pivotal role in medicine by producing medicinally important compounds [105]. Among the streptomycetes, the genus *Streptomyces* produces the majority of the compounds [106]. It has produced 7600 compounds [107]. However, there may still be more undiscovered bioactive compounds. This is because when *Streptomyces coelicolor*’s genome was analyzed in 2002, genes coding to produce 20 bioactive compounds were discovered, even though the bacterium is known to only produce four antibiotics. Therefore, the challenge of finding other ways of stimulating the expression of the other genes remains [105]. Nonetheless, about 75% of all the bioactive compounds applied in the medical treatment of humans and animals are from *Streptomyces* spp. [108].

#### 2.3.2. *Micromonosporineae*

The suborder *Micromonosporineae* has a single family, *Micromonosporaceae*, containing four genera: *Actinoplanes*, *Dactylosporangium*, *Micromonospora*, and *Pilimela*. Collectively, the genera are often called actinoplanetes. They form an elaborate, highly colored, substrate mycelium that lacks aerial hypha. However, commonly raised above the surface of the substrate mycelium are spore-containing-sporangia. The arrangement and morphology of the spores vary among the genera. For instance, in *Actinoplanes* and *Pilimela*, several spores are contained in cylindrical, spherical, or irregular sporangia. Meanwhile, for *Dactylosporangium*, one to six spores are contained in a finger-like sporangia. *Micromonospora* usually produce single spores, usually in branched sporophores [105].

Actinoplanetes are found in diverse habitats, including the soil, freshwater and sea. These bacteria are a source of bioactive metabolites. The genus *Micromonospora* is a prolific producer of antimicrobials, second only to the *Streptomyces* genus [109]. About 8% of the antibiotics used in medicine are from *Micromonospora* species [110].

#### 2.3.3. *Corynebacterineae*

The suborder *Corynebacterineae* has six families: *Nocardiaceae*, *Gordoniaceae*, *Mycobacteriaceae*, *Dietziaceae*, *Tsukamurellaceae* and *Corynebacteriaceae* [105]. In bioprospection, the family *Nocardiaceae* has shown significance. For example, the two genera (*Nocardia* and *Rhodococcus*) belonging to this family contain species capable of producing bioactive compounds [111,112].

The genera *Nocardia* and *Rhodococcus* are largely strict aerobes. They are distributed widely in terrestrial and aquatic ecosystems, wherein they form substrate hypha that fragment into coccoid fragments and rods. The bacteria also develop aerial mycelium that may form conidia. Most of the bacterial strains are free-living saprophytes. However, some strains of *Nocardia* (e.g., *N*. *asteroids*) can cause nocardiosis, an opportunistic disease in humans and animals [105].

#### 2.3.4. *Streptosporangineae*

One of the families of the *Streptosporangineae* suborder is *Nocardiopsaceae* [113]. The family contains a type genus, *Nocardiopsis* [114]. The type species of the genus is *Nocardiopsis dassonvillei,* and IMRU 509 (DSM 43111, ATCC 23218, JCM 7437) is the type strain [114]. Members of the genus are able to prevail under harsh environmental conditions but are generally found in locales with high salt concentrations [115]. Their survival in such conditions is mediated by secreting different and novel bioactive compounds and extracellular enzymes [115]. Thus, the genus is among the biotechnologically important genera of actinobacteria.

### 2.4. Marine Environment as a Source of Microbes Harbouring Novel Bioactive Metabolites

In the last two decades, the number of newly approved antibiotics has dwindled by 75% [116]. For example, since the year 2000, only 30 novel antibiotics have been issued globally. Of these 30, 16 were synthetic, 2 were natural products, and 12 were natural product derivatives. Forty percent of the 30 antibiotics were from actinomycetes, either as natural products or their derivatives [117]. One of the factors contributing to the decline is the rediscovery of known bioactive compounds from conventional microbial habitats, such as the terrestrial habitat. Terrestrial actinomycetes have been bio-prospected extensively; as a result, 70% of commercial antibiotics are extracted from them. Therefore, the likelihood of finding novel bioactive compounds from terrestrial actinomycetes has decreased [116,118]. Thus, the search for novel bioactive compounds has switched to actinomycetes occupying unexplored or underexploited habitats, such as the marine environment [103].

The marine microbial habitats include sea water and sand, coastal and deep sea sediments, mangrove sediments, and hydrothermal vents [119,120]. These habitats tend to have complex conditions such as low temperatures, salinity, and high pressures [120,121]. Therefore, marine microorganisms may develop novel metabolites (absent in terrestrial microbes) in order to survive in such conditions. As a result, a variety of unique bioactive compounds have been recovered from marine actinomycetes in recent times [119,122].

## 3. Anti-Listerial Compounds from Marine Actinobacteria

In 2006, Saha et al. [123] isolated a putatively novel *Streptomyces* MS1/7 from the marine sediments of the Bay of Bengal, India. The actinobacterium produced 2-Allyoxyphenol (**1**) (Figure 2) (molecular formula: C_9_H_10_O_2_), a synthetic compound obtained as a natural product for the first time by Arumugam et al. [124]. The compound was found to have antimicrobial activity against a panel of 21 bacteria and 3 fungi, with MIC values ranging from 0.2 to 1.75 mg/mL [124]. Specifically, 2-Allyoxyphenol had an MIC value of 0.45 mg/mL against *L*. *monocytogenes* MTCC 657, which was determined by the agar dilution method [124].

Another study in India reported on the isolation of the rare actinobacterium *Nocardiopsis* sp. SCA21 from the marine sediment of Havelock Island, the Andaman Islands, and the Nicobar Islands [125]. A bromophenol derivative (4-bromophenol; Figure 2 and Figure 3), and a phthalate ester (Bis (2-ethylhexyl) phthalate; Figure 3, **3**) were purified for the first time from the fermentation broth of the genus *Nocardiopsis* [125]. When tested using the disc diffusion method and the micro-dilution method, respectively, only 4-bromophenol demonstrated activity against *L. monocytogenes* 13932 (zone of inhibition of 24 ± 0.11 mm and MIC value of 62.5 µg/mL) [125].

*Streptomyces blancoensis* 20733 was purified from the marine sediments of San Miguel, Costa Rica, 2020 [126]. Analysis of the bacterium’s extracts led to the isolation of six novel phenolic lipids, adipostatins E–J (**4**–**9**) (Figure 4). Adipostatins E–G exhibited antagonistic activity against *L*. *monocytogenes* ATCC 19115, with IC50 values of 5.9 µM, 34.2 µM, and 20.3 µM, respectively. Adipostatins H–J showed no activity against the ATCC 19115 strain at the tested concentration [126].

Veratraldehyde (**10**) (Figure 5) was purified for the first time from the marine actinomycete *Streptomyces diastaticus* LC360811 (isolated from sediment samples of the Red Sea coast, Egypt) [127]. The compound was active against the tested gram-positive and gram-negative bacteria. However, no activity was detected against *L. monocytogenes* ATCC 35152 [127].

The marine sediment samples collected in July 2021 from Yongzing Island, China, yielded *Streptomyces* sp. ZYX-F-97 [128]. Four new ansamycin derivatives were then purified from the species’ fermentation (Figure 6) [128]. These are 1,19-epithio-geldanamycin A (**11**), 17-demethoxylherbimycin H (**12**), herbimycin M (**13**), and seco-geldanamycin B (**14**). Additionally, eight known ansamycin analogues were isolated (Figure 6) [128]: hervimycin H (**15**), herbimycin I (**16**), tetracyclic thiazinogeldanamycin (**17**), 4.5-dihydro-17-O-demethylgeldanamycin (**18**), 4.5-dihydro-17-S-demethylgeldanamycin (**19**), 19-S-methylgeldanamycin (**20**), 18-methylreblastatin (**21**), and seco-geldanamycin (**22**). But only compounds 14, 18, and 22 showed remarkable inhibition against *L*. *monocytogenes* ATCC 1911, with MIC values of 64 μg/mL, 8 μg/mL, and 16 μg/mL, respectively [128].

Another novel antibiotic, a 22-membered macrolide lactam named haneummycin (**23**) (Figure 7), was purified from *Streptomyces* sp. KM77-8; this was isolated from marine sediment collected from Tokyo Bay, Japan [129]. The antibiotic was then assayed against *Listeria innocua* ATCC 33090^T^ and displayed no anti-Listerial activity at the tested concentration [129].

In summary, out of the 23 reported compounds, only eight of them possessed anti-Listerial activity. The eight compounds and their wide range of other biological activities are consolidated in Table 2. According to Arumugam et al. [124], the hydroxyl and allyloxy groups of compound **1** were responsible for its antimicrobial and antioxidant activities. Due to the absence of potential carcinogenicity and hemolytic toxicity, the cytotoxicity literature suggests the possible application of compound **1** as a food preservative and oral disinfectant [124].

Compound **2** had potent antibacterial activity against a panel of test clinical pathogens, with an MIC value ranging from 7.81 to 125 μg/mL [125]. The compound also showed remarkable free radical scavenging potential, with an IC_50_ value of 187.31 μg/mL against 2,2-diphenyl-1-picrylhydrazyl radicals and an IC_50_ value of 102.22 μg/mL against 2,2′-azinobis (3-ethylbenzothiazoline-6-sulfonic acid) radicals [125]. The iron metal chelating activity of compound **2** was less significant, with an IC_50_ value of >250 μg/mL. Furthermore, compound **2** exhibited remarkable enzyme inhibitory activities against α-glucosidase and α-amylase [125].

Adipostatins exhibited significant antibacterial activity against gram-positive bacteria through the inhibition of co-enzyme A biosynthesis [126]. None of the adipostatins possessed noticeable inhibition against gram-negative microbes (viz., *E. coli*, *Salmonella enterica*, and *Shigella flexneri*). It was suspected that by inhibiting CoA biosynthesis, the adipostatins might also be affecting the synthesis of peptidoglycans and fatty acids. Consequently, the compounds were more potent against gram-positive bacteria [126].

Similarly, the ansamycins (**14**, **18**, **22**) were reported to exhibit significant activity against a panel of gram-positive bacteria [128], but the compounds showed no activity against the gram-negative bacteria (*E. coli* and *Pseudomonas aruginosa*) [128].

## 4. Challenges and Opportunities Associated with Antimicrobials from Marine Actinobacteria

Our literature survey from 1990 to December 2023 reveals the scarcity of published assays testing the anti-Listerial capacity of bioactive compounds purified from marine actinobacteria during this period. For example, only one compound (2-Allyoxyphenol (**1**)) was tested for antibacterial activity against *L*. *monocytogenes* out of the 33 reported antibacterial compounds obtained from marine actinomycetes between 2005 and 2010 in a review by Subramani and Aalbersberg [15]. Furthermore, a review article by Schinke et al. [130] shows that the bioactive compounds derived from marine actinobacteria between 2010 and 2015 were not evaluated for anti-Listerial activity. However, it should be noted that the review only examined bioactive substances with MICs of less than or equal to 20 µg/mL [130]. According to Wang et al. [131], out of the 308 compounds derived from marine actinomycetes from 1990 to 2019, none of them were tested for anti-Listerial activity. Nevertheless, it is not apparent whether this review included studies from the entire year, because in the same year (2019), Siddharth and Rai [125] tested two compounds (4-bromophenol, 2, and bis (2-ethylhexyl) phthalate, **3**) against *L. monocytogenes* 13932 that were isolated from *Nocardiopsis* sp. SCA21. Likewise, from 1992 to 2020, none of the 127 novel halogenated compounds obtained from marine actinomycetes were examined for their anti-Listerial activities [132]. Only *Streptomyces blancoensis* strain 20733 was examined for the biosynthesis of new anti-Listerial compounds (**4**–**9**) out of more than nine marine streptomycete isolates assessed for the production of novel bioactive compounds in 2020 [16]. A literature review (from 2012 to 2022) by Liang et al. [133] shows that around 62 bioactive secondary metabolites were isolated from marine sponge-associated actinomycetes. Interestingly, according to the observed literature, none of the metabolites were examined for their anti-Listerial potential [133]. During these time periods, it appears that the antimicrobial activity assays used were mostly against microbes such as methicillin-resistant *Staphylococcus aureus* (MRSA), vancomycin-resistant *Enterococcus faecium* (VREF), *Bacillus subtilis*, *Candida* species, and others [15,130,134]. This is probably because the World Health Organization considers these bacteria to be the most dangerous group on its list of priority pathogens due to their link to multi-drug resistant nosocomial infections [135].

The research focus for novel anti-Listerial substances appears to be natural antimicrobial agents such as lactoperoxidase, lactoferrin, and lysozyme from animal sources, essential oils and herbal extracts from plant sources, and bacteriocin from microbial sources [136,137,138,139]. These chemicals are primarily being researched for their potential use as natural preservatives to prevent food spoilage and contamination, as *L. monocytogenes* is a major foodborne pathogen [139,140]. These innovative preservatives would aid in the prevention of *L. monocytogenes* growth in foods, while new anti-Listerial drug leads derived from marine actinobacteria could be developed and used to treat infected individuals. Thus, both research approaches are valuable. 

The majority of the published literature is on preliminary screening assays for anti-Listerial compound production by marine actinobacteria that require further analysis to obtain pure compounds (Table 3 and Table 4). For example, researchers [141,142,143,144] only used the cross-streak method to evaluate the anti-Listerial potential of various marine actinobacteria isolated from marine samples collected between 2013 and 2015 (Table 3). Eythorsdottir et al. [145] employed both the cross-streak and agar well diffusion methods to assess the anti-Listerial activity of marine actinobacterial symbionts recovered from shallow-water hydrothermal vents in Northern Iceland (Table 3). During the years 2016 to 2017, different fish were caught from the offshore waters of Ireland, Iceland, and international waters at depths of 1000 m and 850 m [146]. *Arthrobactor* sp. APC 3897 was then isolated from the skin of the deep-sea fish and tested for antibacterial activity by colony overlay assays [147]. The isolate showed activity against foodborne *L. innocua* and *L. monocytogenes* strains [147].

The in vitro anti-Listerial activities of ethyl acetate crude extracts of *Streptomyces cyaneofuscatus* isolates were determined by the disc diffusion method (Table 4) [148]. The agar well diffusion method was used to investigate the anti-Listerial capacity of ethyl acetate crude extracts of *Nocardiopsis alba* isolates (Table 4) [149].

The n-butanol crude extract of *Streptomyces* sp. Sp1 was evaluated for its anti-Listerial activity by the agar well diffusion method (Table 4) [150]. Ngema et al. [85] employed the 96-well microdilution and in silico molecular docking methods to examine the anti-Listerial potential of *Nocardiopsis dassonvillei* SOD(B)ST2SA2′s crude extract (Table 4). Such preliminary screening assays are a yet-to-be-tapped potential source of new anti-Listerial compounds, provided they are processed further. However, researchers may stop processing these crude extracts if the dereplication process fails to reveal novel compounds from the extracts [151].

**Table 3 antibiotics-13-00362-t003:** Preliminary assays of marine actinobacterial activity against *L. monocytogenes* (1990–December 2023).

Marine Actinomycete Strain	Year of Isolation	Country	Method Used for Antibacterial Activity Assay	Test Strain	Reference
*Pseudonocardia carboxydivorans* VO36-3	2013	Chile	Cross-streak method	*L. monocytogenes* 07PF0776	[141]
*Salinoactinospora qingdaoensis* VN6-2					
*Microbacterium profundi* VP2-3					
*Arthrobacter phenanthrenivorans* VO30-3					
*Aeromicrobium alkaliterrae* V040-3					
*Gordonia bronchialis* VO29-3					
*Isoptericola halotolerans* VP3-3					
*Streptomyces janthinus* VS4-2					
*Streptomyces albogriseolus* VH47-3					
*Streptomyces* sp. H-KF8	2013	Chile	Cross-streak method	*L. monocytogenes* 07PF0776	[144]
*Streptomyces* sp. H-KF8	2013	Chile	Cross-streak method	*L. monocytogenes* 07PF0776	[143]
*Rhodococcus* H-CA8F					
*Micrococcus* H-CD9b					
*Kocuria* H-KB6					
*Curtobacterium* H-ED12					
*Curtobacterium* H-BE10					
*Corynebacterium* H-EH3					
*Brachybacterium* H-CG1					
*Brachybacterium* H-CD1					
*Arthrobacter* H-CA8b					
Actinomycete 111	2013	Iran	Cross-streak method	*L. monocytogenes* ATCC 1298	[142]
Actinomycete 112					
Actinomycete 115					
Actinomycete 117					
Actinomycete 127					
Actinomycete 131					
Actinomycete 135					
Actinomycete 141					
Actinomycete 275					
*Tsukamurella strandjordii* 101-1518	2015	Northern Iceland	Agar diffusion assay and cross-streak method	*L. monocytogenes*	[145]

**Table 4 antibiotics-13-00362-t004:** Anti-Listerial crude extracts from various marine actinobacteria (1990–December 2023).

Marine Actinomycete Strain	Year of Isolation	Country of Isolation	Crude Extract	Pathogen Target	Reference
*Streptomyces cyaneofuscatus* M-157	2013	Spain	Ethyl acetate extract	*L. monocytogenes* 72964 *	[148]
*Streptomyces cyaneofuscatus* M-169					
*Streptomyces cyaneofuscatus* M-192					
*Streptomyces cyaneofuscatus* M-207					
*Streptomyces cyaneofuscatus* M-220					
*Streptomyces cyaneofuscatus* M-231					
*Streptomyces* sp. Sp1	2018	Egypt	n-butanol extract	*L. monocytogenes* 19115	[150]
*Nocardiopsis alba* PB-1	2020	India	Ethyl acetate extract	*L. monocytogenes* ATCC 19112	[149]
*Nocardiopsis alba* PB-3					
*N. dassonvillei* SOD(B)ST2SA2	2021	South Africa	Chloroform extract	*L. monocytogenes* KGEO161	[85]
				*L. monocytogenes* ILemanAP345	
				*L. monocytogenes* ILemanEO299	
				*L. monocytogenes* ILemanER317	
				*L. monocytogenes* ILestanBR361	
				*L. monocytogenes* ILestanBR363	
				*L. monocytogenes* ILestanGP395	
				*L. monocytogenes* ILestanGP400	
				*L. monocytogenes* ATCC 15313	

* Cephalosporins-resistant *L. monocytogenes.*

## 5. Conclusions

Generally, new compounds from marine actinobacteria have rarely been tested for their anti-Listerial potential during the past three decades (1990–December 2023). Many countries have not conducted or published any research on the new anti-Listerial compounds purified from marine actinobacteria during this period. However, Costa Rica, Egypt, Japan, China, and India published data on 23 compounds from marine actinobacteria that were tested for their anti-Listerial potential. Eleven of the compounds (adipostatins E–J, ansamycins [11,12,13,14], and haneummycin) were novel. The other compounds (4-bromophenol and bis (2-ethylhexyl) phthalate) were purified for the first time from the genus *Nocardiopsis*. Similarily, veratraldehyde was extracted for the first time from a marine actinomycete. Moreover, a synthetic compound (2-Allyoxyphenol) was obtained as a natural product for the first time. Out of the 23 reported compounds, only 2-allyoxyphenol, adipostatins E–G, 4-bromophenol, and ansamycins (**14**, **18**, and **22**) were found to possess anti-Listerial activity. Considering the zoonotic nature of *L. monocytogenes*, the relatively high case fatality rate of listeriosis, the development and spread of antimicrobial resistance, and the unique nature of secondary metabolites from marine actinobacteria compared to their terrestrial counterparts, it is paramount to expand bioprospecting studies of bioactive compounds from marine actinobacteria, particularly from underexplored ocean environments on the African continent.

## Figures and Tables

**Figure 1 antibiotics-13-00362-f001:**
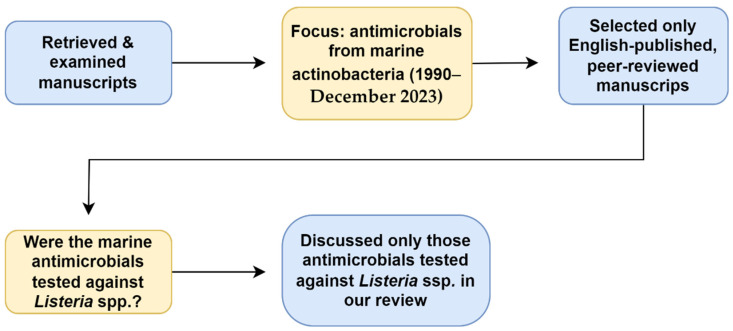
Relevant data retrieval process.

**Figure 2 antibiotics-13-00362-f002:**
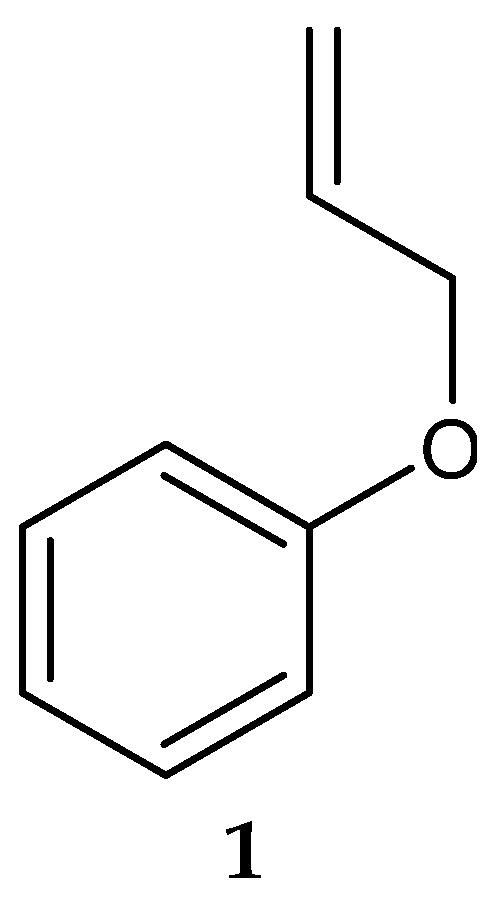
Structure of 2-Allyoxyphenol (**1**) [124].

**Figure 3 antibiotics-13-00362-f003:**
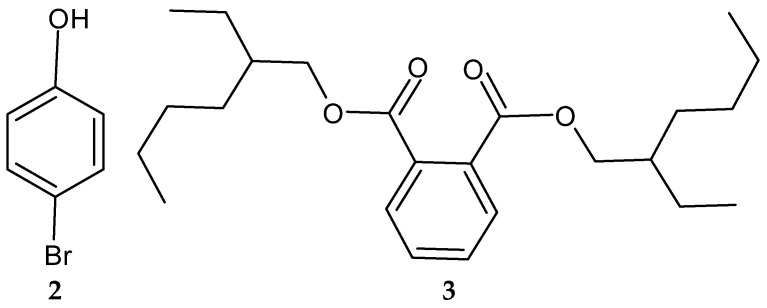
Structures of compounds **2**–**3** [125].

**Figure 4 antibiotics-13-00362-f004:**
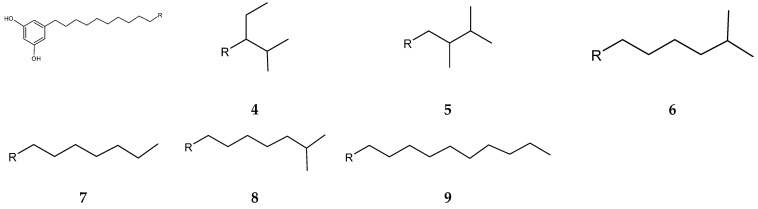
Structures of adipostatins E–J (**4**–**9**) [16,126].

**Figure 5 antibiotics-13-00362-f005:**
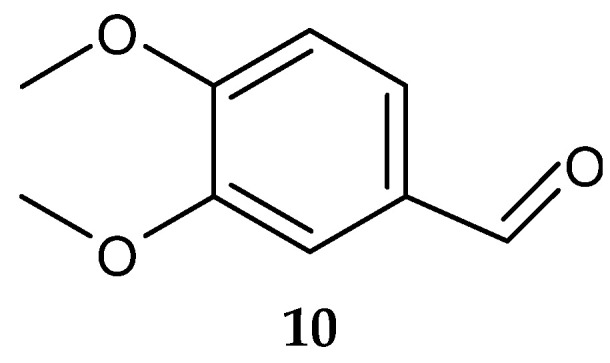
Structure of veratraldehyde (**10**) [127].

**Figure 6 antibiotics-13-00362-f006:**
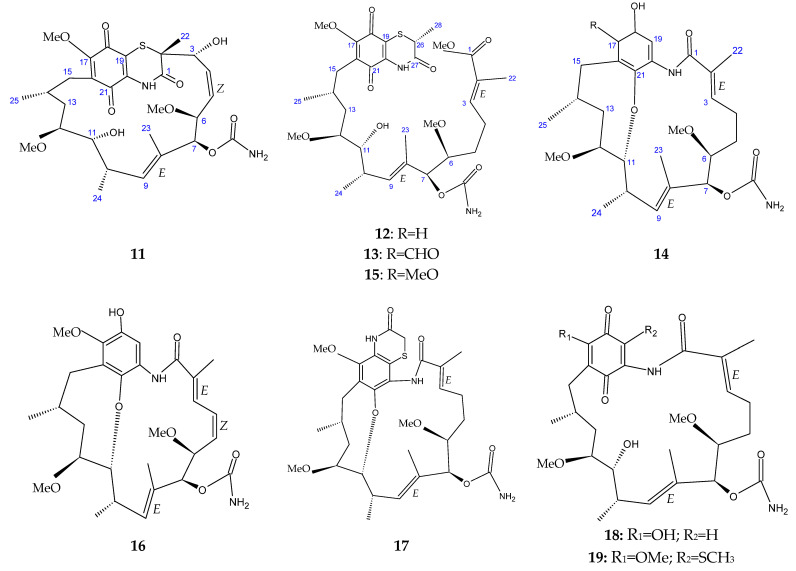
Structures of ansamycins (**11**–**22**) [128].

**Figure 7 antibiotics-13-00362-f007:**
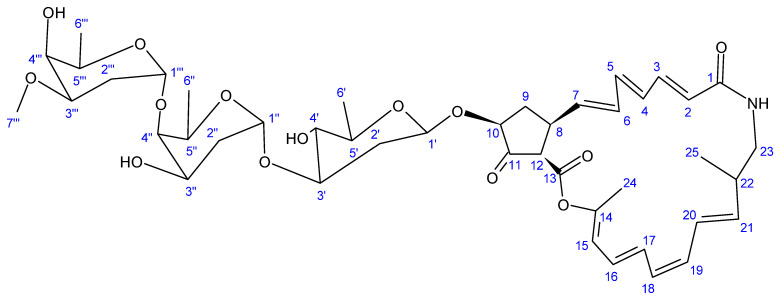
Structure of haneummycin (**23**) [129].

**Table 1 antibiotics-13-00362-t001:** Selected foodborne listeriosis outbreaks from meat and associated products from 1987 to 2019.

Year	Location	No. of Cases (Death)	Type of Meat	Serotype	References
1987–1989	United Kingdom	366 (ND)	Paté	4b	[7]
1900	Australia	9 (6)	Processed meats, paté	—	[7]
1992	France	279 (85)	Pork tongue in jelly	4b	[7]
1993	France	38 (10)	Rillettes	4b	[7]
1996	Australia	5 (1)	Diced, cooked chicken	—	[7]
1998–1999	United States	108 (14)	Hot dogs	4b	[7,31,36]
1999	United States	11 (ND)	Paté	—	[7,29]
1999–2000	France	10 (3)	Rillettes	4b	[7,29]
1999–2000	France	32 (10)	Pork tongue in aspic	4b	[7,29]
2000	United States	30 (7)	RTE deli turkey meat	½a	[7,29]
2000	New Zealand	30 (ND)	RET deli meats	½a	[7,29]
2001	United States	16 (ND)	Deli meats	½a	[7,29]
2002	United States	54 (8)	RET deli turkey meat	4b	[7,29]
2006–2007	Germany	16 (0)	RET scalded sausage	4b	[7,31]
2008	Australia	13 (0)	Jellied pork	4b	[7,37]
2009	Denmark	8 (2)	Beef meat	—	[31,38]
2011	Switzerland	6 (ND)	Cooked ham	½a	[7]
2012	England	14 (0)	Pork pie	—	[31,39]
2012	New Zealand	4 (2)	RTE meat	—	[31,40]
2013	Scotland	3 (0)	Steak pie	½a	[31,41]
2013–2014	Denmark	41 (7)	Meat products	—	[7]
2014	Sweden	51 (0)	Cold cut ham	—	[31,42]
2014	Germany	39 (18)	RTE sausage (pork)	—	[31,43]
2015	Italy	35 (4)	Hog head cheese	½a *	[31,44]
2016	Italy	162 (0)	Cooked beef ham	½a **	[31,45]
2016	Switzerland	7 (0)	Meat pâté (beef)	4b (ST6)	[31,46]
2017–2018	South Africa	1060 (216)	Polony	4b (ST6)	[4,6,7]
2019	The Netherlands	21 (3)	RTE meat products	—	[31,47]

—: no record. * Only 24 of the cases were caused by *L. monocytogenes* serotype ½a. ** All isolates except one were serotype ½a.

**Table 2 antibiotics-13-00362-t002:** Anti-Listerial compounds from marine actinobacteria (1990–December 2023).

Compound	Producing Strain	Biological Activity	References
2-Allyoxyphenol (**1**)	*Streptomyces* MS1/7	Antibacterial; antifungal; antioxidative; food preservative; oral disinfectant	[123,124]
4-Bromophenol (**2**)	*Nocardiopsis* sp. SCA21	Antibacterial; antioxidant; metal chelating; enzyme inhibitory activity	[125]
Adipostatins E–G (**4**–**6**)	*Streptomyces blancoensis* 20733	Antibacterial	[126]
Seco-geldanamycin B (**14**)	*Streptomyces* sp. ZYX-F-97	Antibacterial	[128]
4.5-Dihydro-17-O- demethylgeldanamycin (**18**)			
Seco-geldanamycin (**22**)			

## Data Availability

Not applicable.

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
