# Peer review of "A Mini-Review of Anti-Listerial Compounds from Marine Actinobacteria (1990–2023)"

_antibiotics, 2024, doi:10.3390/antibiotics13040362_

Round 1

Reviewer 1 Report

Comments and Suggestions for Authors

Dear authors,
The document presents important information, however, it lacks an
objective and a methodology that supports the obtaining and analysis of
the bibliography. It is necessary to properly structure the document.

In my opinion a manuscript without methodological foundation should not
be considered for publication since there is no evidence of serious
work, even if the information presented is important. However, I share
the following comments.

Specific comments

The authors must write the objective of the article.
The authors have to mention their search criteria, keywords.
The authors must mention in which databases the searches were carried out.
What were your criteria for including the articles on which the review
was based?

Comments on the Quality of English Language

Moderate editing of English language required

Author Response

[Antibiotics] Manuscript ID: antibiotics-2900585

Dear Editor,

Thank you for the opportunity to revise our review manuscript. The responses to reviewers’ comments are elaborated below.

Responses to Reviewer 1

Reviewer 1 Comments

Dear authors,
The document presents important information, however, it lacks an
objective and a methodology that supports the obtaining and analysis of
the bibliography. It is necessary to properly structure the document.

In my opinion a manuscript without methodological foundation should not
be considered for publication since there is no evidence of serious
work, even if the information presented is important. However, I share
the following comments.

Specific comments

The authors must write the objective of the article.
The authors have to mention their search criteria, keywords.
The authors must mention in which databases the searches were carried out.
What were your criteria for including the articles on which the review
was based?

Author responses: The introduction of the article has been revised to include the objective of the article, the search criteria used to obtain data, and the databases used to carry out the searches.

We thank Reviewer 1 for the insightful comments.

Reviewer 2 Report

Comments and Suggestions for Authors

In this manuscript, the authors summarized anti-Listerial compounds from marine ac-2 tinobacteria. the organization and innovation of the manuscript should be further strengthened. Below comments may help authors revise the manuscript.

1.     Some reviews have been reported for summarizing the sources, chemical structures, and anti-Listerial activities of bioactive compounds derived from marine actinobacteria. What is the novelty of this review?

2.     Furthermore, you addressed Chemotherapeutic treatment for human listeriosis. Have you examined the literature regarding alternative treatments? Additionally, consider incorporating a figure into this section to enhance comprehension.

3.     Include a schematic in this review paper to comprehensively illustrate its aims.

4.     Since the title of the manuscript is "A Mini-Review of anti-Listerial compounds from marine ac-2 tinobacteria (1990–2023)" it is suggested to discuss the precise application of anti-Listerial compounds. It is advised for the authors to introduce a table to summarize the application of anti-Listerial compounds.

5.     Could you please clarify which software you utilized to generate the compound structures? For example, in Figures 2 and 3, the structures appear to be blurry.

6.     In the abstract, it would be beneficial for the authors to emphasize the content covered in the review rather than providing a summary of information.

7.     It would enhance clarity to consolidate the anti-Listerial compounds into a single table.

8.     The introduction part should be revised. Some recent literature in this field should be cited in the introduction parts. See for instance:

https://www.sciencedirect.com/science/article/abs/pii/S0169433223002507

https://link.springer.com/article/10.1007/s13738-019-01595-5

Comments on the Quality of English Language

Moderate editing of English language required.

Author Response

[Antibiotics] Manuscript ID: antibiotics-2900585

Dear Editor,

Thank you for the opportunity to revise our review manuscript. The responses to reviewers’ comments are elaborated below.

Reviewer 2 Comments and Suggestions for Authors

Reviewer comments:

In this manuscript, the authors summarized anti-Listerial compounds from marine ac-2 tinobacteria. the organization and innovation of the manuscript should be further strengthened. Below comments may help authors revise the manuscript.

  1. Some reviews have been reported for summarizing the sources, chemical structures, and anti-Listerial activities of bioactive compounds derived from marine actinobacteria. What is the novelty of this review?

Answer:  The introduction of this review has been revised to explicitly state the novelty of the review

  1. Furthermore, you addressed Chemotherapeutic treatment for human listeriosis. Have you examined the literature regarding alternative treatments? Additionally, consider incorporating a figure into this section to enhance comprehension.

Answer: We thank the reviewer for this observation. The literature regarding alternative treatments are outside the scope of this review, hence they are not included. We therefore included this aspect as a limitation of this review.

  1. Include a schematic in this review paper to comprehensively illustrate its aims.

Answer: The introduction section of this review has been revised to comprehensively state its aims

  1. Since the title of the manuscript is "A Mini-Review of anti-Listerial compounds from marine ac-2 tinobacteria (1990–2023)" it is suggested to discuss the precise application of anti-Listerial compounds. It is advised for the authors to introduce a table to summarize the application of anti-Listerial compounds.

Answer: To address this comment, we included Table 2 in our review manuscript.

  1. Could you please clarify which software you utilized to generate the compound structures? For example, in Figures 2 and 3, the structures appear to be blurry.

Answer: The compound structures have been redrawn using the ChemSketch software

  1. In the abstract, it would be beneficial for the authors to emphasize the content covered in the review rather than providing a summary of information.

Answer: The abstract has been revised to emphasize the content covered in the review.  .

  1. It would enhance clarity to consolidate the anti-Listerial compounds into a single table.

Answer: All the anti-Listerial compounds have been consolidated into Table 2

  1. The introductionpart should be revised. Some recent literature in this field should be cited in the introduction parts. See for instance:

https://www.sciencedirect.com/science/article/abs/pii/S0169433223002507

https://link.springer.com/article/10.1007/s13738-019-01595-5

Answer: the introduction part has been revised to include recent literature

We thank reviewer 2 for the insightful comments.

Round 2

Reviewer 2 Report

Comments and Suggestions for Authors

The manuscript has undergone a comprehensive revision by the author and is now suitable for acceptance in this journal.